# Radiotherapy for Leptomeningeal Carcinomatosis in Breast Cancer Patients: A Narrative Review

**DOI:** 10.3390/cancers14163899

**Published:** 2022-08-12

**Authors:** Ewa Pawłowska, Anna Romanowska, Jacek Jassem

**Affiliations:** Department of Oncology and Radiotherapy, Faculty of Medicine, Medical University of Gdansk, 80-210 Gdansk, Poland

**Keywords:** leptomeningeal carcinomatosis, breast cancer, radiotherapy

## Abstract

**Simple Summary:**

Leptomeningeal carcinomatosis (LC) is a rare event in breast cancer (BC) patients that carries an abysmal prognosis. Little progress has been made in this field in the last few decades. Despite innovations in radiotherapy (RT), there is no univocal evidence of its impact on survival. Due to the rarity of the diagnosis, only a few prospective trials have evaluated the role of RT for LC in BC. Nonetheless, most BC patients with LC currently receive RT, depending on local protocols and individual convictions. This review presents the current knowledge on the indications and feasibility of RT for LC in BC, focusing on new technologies and perspectives.

**Abstract:**

Leptomeningeal carcinomatosis (LC), defined as the infiltration of the leptomeninges by cancer cells, is a rare oncological event with the most common etiology being breast cancer (BC), lung cancer, and melanoma. Despite innovations in radiotherapy (RT), firm evidence of its impact on survival is lacking, and concerns are related to its possible neurotoxicity. Owing to a paucity of data, the optimal treatment strategy for LC remains unknown. This review discusses current approaches, indications, and contraindications for various forms of RT for LC in BC. A separate section is dedicated to new RT techniques, such as proton therapy. We also summarize ongoing clinical trials evaluating the role of RT in patients with LC.

## 1. Introduction

Leptomeningeal carcinomatosis (LC), defined as an infiltration of the leptomeninges by cancer cells, is a rare event in solid tumors, with the most common etiology being breast cancer (BC), lung cancer, and melanoma [1]. The reported incidence of LC in BC patients ranges widely from 0.8% to 6.6% in clinical reports and from 2.6% to 16% in autopsy series [2,3,4,5,6,7,8,9]. However, these statistics can be biased by analyzing cohorts of high-risk patients. Indeed, in an unselected cohort of 1915 BC patients, the 5-year incidence of LC was merely 0.3% [10]. The coexistence of LC with brain metastases (BM) has been reported in 4–14% of cases, but the actual rates remain unknown [11,12,13]. The incidence of LC in BC is higher in younger patients and those with a larger primary tumor, advanced nodal disease, histological grade 3, negative estrogen receptor (ER), positive human epidermal growth factor receptor 2 (HER2), triple-negative type, and a high proliferative index [14]. Lobular carcinoma is particularly associated with LC and accounts for about 35% of BC cases [14]. This is in contrast with only about a 7% incidence of lobular carcinoma among parenchymal BM, suggesting an affinity of this subtype for leptomeningeal dissemination.

Previous surgery for BM, especially for supratentorial tumors, also increases the risk of LC [15,16]. A meta-analysis showed that BM resection followed by stereotactic radiosurgery (SRS) carries a higher risk of developing LC than resection followed by whole-brain radiotherapy (WBRT) [17]. Another suggested risk factor for developing LC is SRS used alone [18,19,20]. However, in a large retrospective study including nearly 500 patients administered SRS alone or surgery followed by SRS, surgical resection was the sole predictor of LC risk [15]. In a retrospective analysis by Trifiletti et al., LC after SRS for BM occurred in only 9% of patients 12 months after diagnosis of BM, and active lung metastases at the time of SRS worsened the prognosis [21]. Nevertheless, the incidence of LC in BC is rising owing to longer survival, poor blood–brain barrier permeability of most medications, and better LC detection with novel neuroimaging techniques [22].

The diagnosis of LC is difficult due to the variability and non-specificity of symptoms. Clinical manifestations include headache, neurological deficits, radicular pain, cauda equina syndrome, sensory disturbances, seizures, somnolence, nausea and vomiting, and psychiatric disorders [14]. About 80% of patients are symptomatic at diagnosis, with headache being the most common ailment [23]. According to joint recommendations of the European Association of Neuro-Oncology (EANO) and the European Society for Medical Oncology (ESMO), diagnostic criteria determining the likelihood of LC diagnosis as “confirmed,” “probable,” “possible,” or “no evidence for” include clinical symptoms, craniospinal magnetic resonance (MRI), cerebrospinal fluid (CSF) examination, and focal biopsy [24].

Patients with BC LC have a dismal prognosis, with a median overall survival (OS) ranging from four to six weeks without treatment to six months with intensive multimodal treatment [25,26]. The reported one-year survival rate is approximately 20% [27,28]. The individual OS may be estimated using prognostic indexes. A recently suggested INDEX score includes age, performance status (PS), BC subtype, and treatment intensity [29]. The scoring system proposed by Gauthier et al. contains PS, hormone receptor status, number of chemotherapy (ChT) regimens before LC diagnosis, and Cyfra 21-1 level in CSF [30].

The main aim of treating BC LC is to prolong survival with an acceptable quality of life, particularly by preventing or delaying neurological deterioration [24,31,32]. However, the optimal therapeutic management of this entity has not been established, as there are no data from randomized trials. Hence, treatment guidelines are currently based on expert opinions or clinical experience. There is a paucity of data on radiotherapy (RT) use in BC LC. Nevertheless, in clinical practice, 13.6–80% of BC patients with LC receive RT [14].

This review discusses the indications and limitations of RT in BC patients with LC, implementations of new RT techniques, and challenges in this field.

## 2. Materials and Methods

PubMed, ClinicalTrials.gov, and Google Scholar were searched in December 2021. The following keywords or combinations of them were used: “leptomeningeal disease,” “leptomeningeal carcinomatosis,” “metastases,” “radiotherapy,” “irradiation,” “radiation therapy,” “breast cancer,” “guidelines,” “original article,” “original paper,” and “review.” Articles in languages other than English were excluded. The reference lists of the retrieved articles were checked to detect other articles that might be of interest to this narrative review.

## 3. Results

### 3.1. Impact of Radiotherapy on Survival

The survival benefit of RT in BC LC is still debatable, as the current knowledge is based on non-randomized studies (Table 1). Furthermore, the results of particular studies are conflicting, even among single research teams [29,33,34]. 

Two prospective trials evaluating the role of combined intrathecal chemotherapy (ITC) and RT showed the OS benefit of RT [36,37]. However, both included patients with LC from various solid tumors, with only a small subset of BC patients. In the phase 2 study, patients with at least one adverse prognostic factor (Karnofsky performance status [KPS] of < 60%, severe and multiple neurological deficits, encephalopathy, extensive systemic disease with few treatment options, and bulky BM) received MTX ITC concurrently with RT [37]. The concomitant treatment was well tolerated, with no major toxicities or side effects related to RT. Mild or moderate skin reactions and hair loss occurred in all patients undergoing brain RT, and 22% experienced mild and moderate otitis media. Moderate and severe toxicity occurred in 20% of cases, which seems acceptable, considering the expected OS benefit. In the second trial, comparing single and combination ITC, RT to the brain, spine, or whole craniospinal axis was administered in 50% of cases [36]. Concurrent RT significantly improved the response rate and OS; however, allocation to RT was not randomized.

As several studies have shown that CSF flow interruption is associated with decreased survival, RT remains the treatment of choice to remove the flow obstruction, reduce toxicity, and enhance the efficacy of ITC [38,39,40,41].

### 3.2. Whole-Brain Radiotherapy

WBRT is still the most widely used RT technique in LC treatment [40]. However, assessment of its impact on survival as a single modality is difficult, as in most cases, it is combined with systemic or intrathecal ChT (Table 2).

In a retrospective analysis investigating the efficacy of WBRT as a single modality, conventionally fractionated RT was performed via parallel opposed fields [43]. The planning target volume encompassed the whole brain and the meningeal space (i.e., the lamina cribrosa and basal cisterns). The toxicity of RT was low, with alopecia, nausea, headache, and fatigue being the most common side effects. There was no grade 3 or 4 toxicity. The authors concluded that WBRT is an effective palliative treatment of LC for patients unfit for ChT and with low KPS. Nevertheless, improvement of neurological deficits was reported in only 11% of patients. The safety of WBRT was also confirmed in a prospective randomized study assessing the role of ITC in LC [44]. RT did not increase neurotoxicity, even if combined with ITC. However, in a historical series, the same author described disseminated necrotizing leukoencephalopathy (DNL) in four patients with BC LC treated with WBRT, followed by low-dose ITC MTX [45]. As DNL also developed in five non-irradiated patients, the results were inconclusive.

According to the German Society of Radiation Oncology (DEGRO) guidelines, the clinical target volume in WBRT should encompass the cerebrum plus cerebellum and the brainstem down to the caudal limit of the second vertebral body [32]. Importantly, the meningeal space with the lamina cribrosa and basal cisterns should be included. Preferred dose regimens are 30 Gy/10 fx (5 fx per week) and 20 Gy/5 fx in patients with an unfavorable prognosis or 20 Gy/10 fx in patients with a predicted survival exceeding 12 months.

In a single-center retrospective study, Okada et al. showed that the dose of 30 Gy given in ≥10 fx provided significantly better OS than 30 Gy in <10 fx (median OS of 2.6 and 0.6 months, respectively); however, the patient groups were small (24 and seven, respectively) [46].

### 3.3. Stereotactic Body Radiation Therapy

According to the EANO-ESMO and National Comprehensive Cancer Network (NCCN) guidelines, focal RT should be considered for well-circumscribed, symptomatic lesions. It can relieve cauda equina syndrome, cranial palsies, focal pain, or obstruction of the CSF flow in 30% and 50% of patients with spinal and intracranial blocks, respectively [24,47]. Thanks to the high precision of treatment delivery and higher biologically effective dose, SRS may be a preferred treatment option for central nervous system (CNS) lesions localized near critical structures [48]. DEGRO guidelines for palliative RT in metastatic BC propose SRS at a single dose of 15–25 Gy (specified for isodose 80–90%) for lesions smaller than 3.5 cm in diameter and fractionated stereotactic RT for bigger lesions. Depending on the treatment volume, recommended fractionation schedules are 4 × 8.7 Gy, 5 × 7 Gy, 6 × 5 Gy, or 10 × 4 Gy. In the case of additional WBRT, the single fraction of 15–18 Gy (depending on the tumor size) or fractionated regimen of 6 × 5 Gy are preferred. The gross tumor volume is delineated at the MRI, and the planning target volume is created by adding an isotropic margin of 1–2 mm [31,32].

We have not identified any phase 2 or 3 randomized studies of SRS for LC in BC patients. The recommended RT regimens are extrapolated mainly from BM treatment or based on retrospective reviews, case series, and expert opinions. Most of the studies included patients of different histologies or evaluated mixed SRS/WBRT cohorts. Nevertheless, considering the potential benefits, this option seems reasonable whenever focal irradiation is indicated.

In the series by Wolf et al., out of 16 patients with LC managed with cranial SRS, five were BC patients [49]. In the entire group, five patients had received WBRT earlier. The median margin dose delivered was 16 Gy in a single fraction of the 50–80% isodose volumes. Subsequent MRI was available for 14 patients. Five achieved disease stabilization, eight partial remissions, and one progressed. The median OS from the end of SRS for LC was 10 months, and the one-year OS was 26%. Six more patients needed subsequent WBRT due to distal progression, with a median gap of six months since SRS. The authors concluded that focal LC could be successfully performed with SRS. In some cases, SRS can eliminate or postpone WBRT with its side effects, including neurocognitive dysfunctions, alopecia, and fatigue [49]. Lekovic et al. described a case of a BC patient treated with a combination of SRS, craniospinal irradiation (CSI), and ITC with trastuzumab [50]. During the course of the disease, she received 24 Gy/3 fx for Meckel’s cavity and auditory canal tumors, CSI of 30 Gy with ITC, followed by focal RT to spinal metastases (25 Gy/5 fx) and the cerebellar hemisphere (18 Gy/1 fr.). This multimodal treatment allowed for an impressive 46-month good-quality survival.

### 3.4. Proton Therapy

Proton therapy (PT), with its unique physics resulting in a steep dose decrease, is a very tempting option for CNS treatment, particularly for CSI. A classic photon RT results in a substantial dose delivered to the whole spinal column and anteriorly located organs, mainly the intestines and kidneys, and is rarely used [31,51,52,53]. Proton CSI allows for less gastrointestinal and hematological exposure [54]. PT of CSI was investigated in a prospective dose-escalation phase I trial including 21 patients with LC from solid tumors, seven of whom with BC [55]. The clinical target volume included the entire brain, with proper coverage of the meninges, thecal sac, and proximal sacral nerve roots. As there was no dose-limiting toxicity (DLT) in the first six patients, all subjects received a 30 Gy relative biological effectiveness (RBE) dose in 3 Gy RBE/fx. This hypofractionated regimen replicated a popular palliative photon beam RT schedule. DLT, including grade 4 lymphopenia, grade 4 thrombocytopenia, and grade 3 fatigue, occurred in two patients from the expansion cohort and resolved without any specific treatment. The median OS was 8 months, and four patients achieved CNS disease control for longer than 12 months. The authors concluded that hypofractionated proton CSI is safe and feasible in patients with LC.

Most recently, a randomized Phase II study compared PT of CSI with standard involved-field photon RT in 63 patients with LC (36 patients with non-small cell lung cancer and 27 with breast cancer) [56]. The CNS-PFS favored PT (median 7.5 months vs. 2.3 months with photon-beam RT; *p* < 0.001). OS was also superior with PT (median 9.9 months vs 6.0 months for photons; *p* = 0.029). There were no significant differences between both therapies in the frequency of grade 3 and 4 toxicities.

### 3.5. Craniospinal Irradiation

Due to the presence of cancer cells in the CSF, the neuroaxis seems to be a reasonable target in LC. However, the use of photon CSI is discouraged by international guidelines due to its significant toxicity, the difficulty of RT planning, and its unconfirmed survival benefit [23,24,31,47]. So far, no trials have evaluated the feasibility of CSI exclusively in the BC population, and all knowledge is based on small case series and reviews (Table 3).

Two studies reported the toxicity of CSI with 2D planning [51,52]. In the study by Hermann et al., patients were treated with CSI with (*n* = 10) or without (*n* = 9) ITC MTX [52]. Early adverse events included myelosuppression (G3 in four patients and G4 in one), dysphagia, mucositis, and nausea. There was no late toxicity. In another study, 17 symptomatic patients (six with BC) were irradiated for the entire neuraxis, with an additional WBRT dose of up to 50.4 Gy in nine patients and concomitant ITC MTX in five [52]. There was one toxic death due to an intracranial hemorrhage. Late toxicities included grade 3 infection in one patient and grade 1 myelitis in three (18%). Eleven patients received further therapies after CSI.

The excessive toxicity of CSI described in the aforementioned studies may likely be limited with modern RT techniques, such as volumetric modulated arc therapy (VMAT), helical tomotherapy (HTT), or PT. In a case report of a BC patient with LC treated with VMAT CSI, the mean bone marrow dose was 15.3 Gy, and bone marrow V20 was only 36% [59]. HTT, evaluated in three studies, was also found to be a useful therapeutic modality with acceptable toxicity [53,57,58]. However, one study reported a worrisome incidence of serious adverse events, including three toxic deaths [58].

Some authors have attempted to develop prognostic scores for decision-making. In one study, age below 55 years, KPS > 70%, and neurological response to treatment were identified as favorable prognostic factors for OS [31]. In another study, risk factors included KPS < 70% and the coexistence of an extracranial disease [57]. The median OS for patients with no, one, and two risk factors was 7.3 mo., 3.3 mo., and 1.5 mo., respectively.

A recently published review summarized 13 studies, including a total of 275 patients treated with CSI for LC of different histologies (the most common being leukemia and BC) [60]. The median CSI dose was 30 Gy, and 18% of patients received PT. Fifty-two percent of patients had improvement or stabilization of neurological symptoms. The median OS for all patients and for those managed with marrow-sparing PT was 5.3 months and 8 months, respectively. The most common treatment-related toxicities were fatigue and hematologic and gastrointestinal events. The authors concluded that CSI is a viable, yet relatively toxic option for LC. Proton CSI was discussed in an earlier section.

### 3.6. Radiotherapy Guidelines

The current RT guidelines for LC are summarized in Table 4. The NCCN guidelines stratify patients with LC into two categories [47]. The good-prognosis group consists of those with KPS ≥ 60%, no major neurologic deficits, minimal systemic disease, and availability for reasonable systemic treatment options. Patients with KPS < 60%, multiple, serious, major neurologic deficits, extensive systemic disease, limited treatment options, bulky CNS disease, and encephalopathy are a poor prognostic group. For the group with a good prognosis, NCCN recommends systemic ChT, ITC, or RT. Patients in the poor prognosis group may receive palliative treatment or the best supportive care. NCCN guidelines do not specify the technical aspects of RT, such as the dose or irradiated volume, which should depend on the histology and sites requiring palliation.

The EANO-ESMO guidelines recommend, in general, treatment of LC with ChT or ITC, targeted therapies, and RT, or their combination [24]. Typical target volumes for RT in patients with cranial neuropathies include the skull base, the interpeduncular cistern, and the first two cervical vertebrae. In patients with cauda equina syndrome, the irradiated volume should include the lumbosacral vertebrae. Guidelines allow for focal irradiation for cauda equina syndrome or cranial nerve palsies after excluding other causes, even without corresponding MRI findings.

In 2010, DEGRO published practical guidelines for palliative RT of BM and LC in BC patients [31,32]. In LC with spinal manifestation, the clinical target volume should encompass the gross tumor volume with a safety margin matched to the individual clinical requirements. The guidelines also mention the RT technique and dose schemes cited in the appropriate sections.

### 3.7. Future Perspectives

Since LC is a manifestation of disease spread, it is unlikely that any radiotherapy developments will substantially improve the survival of this miserable entity. However, newer RT techniques, such as PT, SRS, HTT, or heavy ion irradiation, can decrease treatment toxicity. The main issue of LC systemic treatment is the poor blood–brain barrier permeability of most medications. The activity of a paclitaxel trevatide, a new experimental drug designed to have greater potential to cross this barrier, will be assessed in a phase 3 trial (NCT03613181). Recently, an increasing number of BC patients with targetable molecular alterations have been managed with targeted therapies. In advanced HER2-positive BC, the combination of trastuzumab, pertuzumab, and docetaxel is considered the standard first-line treatment [61]. A recent meta-analysis showed that IT trastuzumab is a reasonable and safe treatment for BC LC. This method resulted in CNS-PFS of 5.2 months, a median OS of 13.2 months, and a significant clinical improvement in 55% of cases [62]. In the NCT04588545 trial, patients with LC will receive this regimen intrathecally, concurrently with WBRT or focal RT. Low molecular weight HER2-tyrosine kinase inhibitors, such as lapatinib, neratinib, and tucatinib, have been shown to be effective in BC patients with BM, but there are scarce data on their activity in LC [63,64]. A phase 1 study NCT03661424 will evaluate the role of a bi-specific antibody (HER2Bi)-armed activated T-cells (HER2 BATs) in HER2 positive patients with meningeal spread. BC patients harboring hereditary *BRCA1/2* mutations respond to treatment with polyadenosine diphosphate ribose polymerase inhibitors, including olaparib, veliparib, talazoparib, and iniparib, and these compounds are other potential options in LC treatment [65]. Inhibition of the cyclin D1 pathway (CDK4/6 inhibitors) is an effective strategy for ER-positive BC, but its clinical efficacy in LC from BC is disappointing [66,67]. For tumors not harboring drug-targetable mutations, another option is systemic immunotherapy, especially checkpoint blockade with antibodies against the programmed cell protein-1 (PD-1) or its ligand (PD-L1). In a phase 2 study using pembrolizumab (PD1-antibody), 60% of 20 patients (17 with BC) met the primary endpoint of three-month OS [68]. Toxicities of grade 3 and higher, most frequently hyperglycemia, nausea, and vomiting occurred in 40% of the patients. A recruiting NCT03719768 trial will evaluate the role of concurrent RT and avelumab, a PD-L1 antibody, on BC LC.

## 4. Discussion

LC is a rare and devastating event in the course of BC. The survival of patients treated for LC from BC has not significantly improved within the past few decades. Our PubMed.gov search identified more than 280 articles, with a growing number of publications in recent years (Figure 1). However, only 39 (~14%) are original papers, with single publications annually (Figure 2).

The optimal management of LC remains undefined, as there is no level I evidence from randomized clinical trials. RT, used alone or combined with systemic or intrathecal therapies, remains the main treatment modality for LC. However, its use is based on standard practices, local protocols, or individual presumptions, and not on robust evidence. The only recent recommendations for the treatment of LC come from NCCN and are dedicated to LC in general, and not specifically to BC. The DEGRO and EANO-ESMO guidelines were published 12 and 5 years ago, respectively. Due to the rarity of LC and its dismal prognosis, only a few prospective trials have been conducted. All were phase 1 or 2 (Table 5). Symptomatically, only one out of four completed trials published its final results. As of April 2022, there are six ongoing trials. Of those, four are evaluating the role of RT combined with intrathecal or systemic therapy, one is using proton RT alone for CSI irradiation, and one is comparing proton CSI versus photon IF-RT, including WBRT, focal spine RT, or their combinations. Only one of these trials is dedicated to BC patients; the remaining include mixed populations.

Another issue is the objective assessment of the response to RT in LC. The response evaluation criteria in solid tumors (RECIST) are not useful, as the infiltration of the meninges is often not measurable with this instrument. Numerous studies have shown that CSF cytology does not correlate with survival and clinical response, likely due to false-negative testing of CSF [37]. Consequently, most studies used clinical evaluation based on neurological examination. However, the methodology of clinical assessment is subjective, may not be reproducible, and does not apply to all patients with LC (e.g., to patients with cognitive disorders). The Leptomeningeal Assessment in Neuro-Oncology (LANO) group developed a dedicated tool for evaluating the treatment response in LC [69]. However, due to its complexity and problems with validation, it has not been routinely implemented. An updated, simplified version of the LANO scorecard is under evaluation [70]. According to the EANO-ESMO recommendations, the diagnosis, response assessment, and follow-up of LC in BC patients should be based on a complete neurological examination, neuroimaging evaluation, and CSF cytology [24]. This classification seemed to be highly prognostic and was recommended for the stratification and design of clinical trials [71].

## 5. Conclusions

LC is a rare event in breast cancer patients and carries a bleak prognosis. Despite innovations in RT, little progress has been made on the use of this method in LC. Due to the rarity of the diagnosis, only single prospective trials have evaluated the role of RT for LC from BC. Faced with the difficulties in conducting prospective clinical trials, a registry of BC patients with LC might shed more light on this disastrous entity.

## Figures and Tables

**Figure 1 cancers-14-03899-f001:**
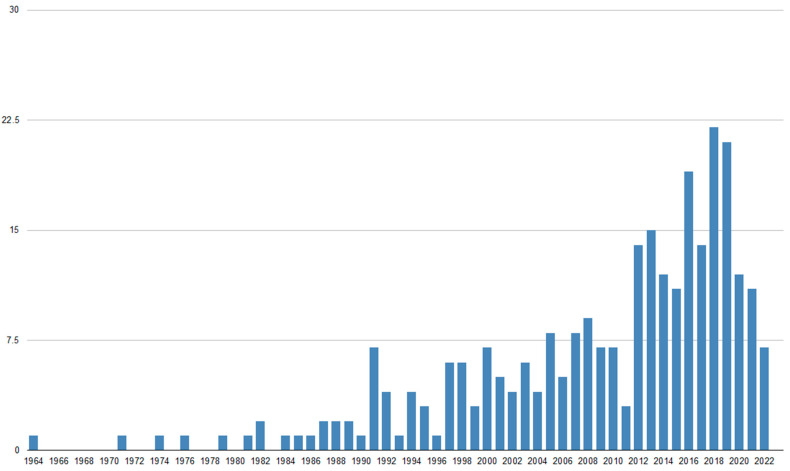
The number of publications per year on radiotherapy for leptomeningeal carcinomatosis in breast cancer (Pubmed.gov database).

**Figure 2 cancers-14-03899-f002:**
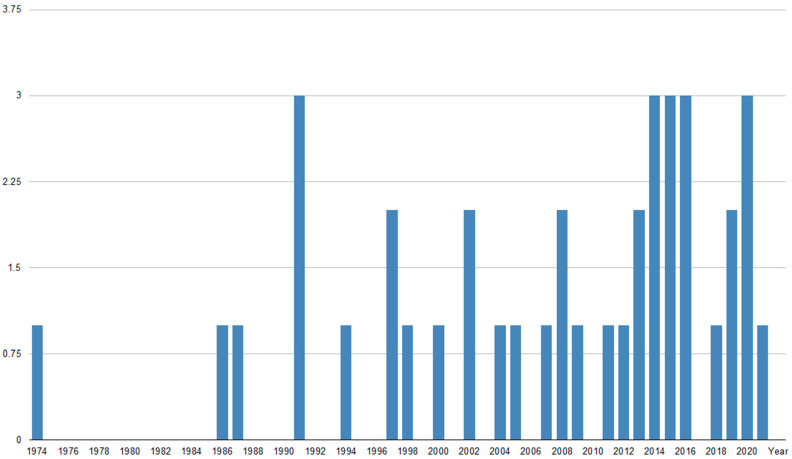
The number of original publications per year on radiotherapy for leptomeningeal carcinomatosis in breast cancer (Pubmed.gov database).

**Table 1 cancers-14-03899-t001:** Clinical studies evaluating the survival impact of radiotherapy for leptomeningeal carcinomatosis in breast cancer.

Authors	Number of Patients (BC)	Study Type	Treatment	Major Results	Toxicity (Including All Treatment Methods)
Niwińska et al. [33]	118 (118)	Prospective	Treatment of physicians’ choice ChT-68% ITC-79% WBRT-56% spinal cord RT-24%	Brain RT-prolongs survival in univariate analysis (*p* = 0.017), not confirmed in multivariate analysis (*p* = 0.817); No OS benefit of spinal cord RT (*p* = 0.894)	
Rudnicka et al. [34]	67 (67)	Prospective	Treatment of physicians’ choice ITC 85% ChT 61% WBRT 49% Spinal cord RT 15%	Brain RT-prolongs survival in univariate analysis (*p* = 0.004), not confirmed in multivariate analysis (*p* = 0.156); No OS benefit of spinal cord RT (*p* = 0.989)	
Niwińska et al. [29]	187 (187)	Prospective	Treatment of physicians’ choice ITC 68% ChT 56% WBRT 35% spinal cord RT 8% WBRT + spinal cord RT 13%	Multivariate analysis: RT improves survival (*p* < 0.001)	
Kingston et al. [35]	182 (182)	Retrospective	Treatment of physicians’ choice ITC 7.7% ChT 25% RT 34% best supportive care 20.3%	Longer OS (median 6.1 mo.) and PFS (median 5.8 mo.) with RT compared to ITC or palliative care alone	
Hitchins et al. [36]	44 (11)	Prospective, randomized	Arm A: ITC MTX Arm B: ITC MTX + Ara-C ChT 68% RT 50% (WBRT *n* = 17, spinal RT *n* = 4, neuroaxis *n* = 1)	Improved OS with concurrent ITC and WBRT (*p* = 0.003) compared to ITC alone RR 73% and 35%, with and without RT, respectively, (*p* < 0.05) Median OS of 4 mo. and 1.8 mo., with and without RT, respectively	Nausea and vomiting: 45% Meningitis: 14% Septicemia, neutropenia: 12% Mucositis: 12% Pancytopenia: 10%
Pan et al. [37]	59 (11)	Prospective, single-arm	Induction, concomitant and consolidation ITC (MTX) + IF-RT (40–50 Gy/20 fx)	Univariate analysis: longer OS in patients achieving clinical response (*p* = 0.013) and administered a complete course of concomitant (ITC + RT) therapy (*p* = 0.016)	Acute cerebral meningitis: 2% Chronic encephalopathy: 5% Radiculitis: 27% Bone marrow depression: 22% Mucositis: 20% Leukodystrophy: 68% Encephalopathy: 19%

BC, breast cancer; ChT, chemotherapy; ITC, intrathecal chemotherapy; WBRT, whole-brain radiotherapy; RT, radiotherapy; OS, overall survival; PFS, progression-free survival; mo., months; MTX, methotrexate; Ara-C, cytosine arabinoside; RR, response rate; IF-RT, involved-field radiotherapy; Gy, Gray; fx, fractions.

**Table 2 cancers-14-03899-t002:** Clinical studies evaluating the role of whole-brain irradiation for leptomeningeal carcinomatosis in breast cancer.

Authors	Study Type	Number of patients (BC)	RT Dose	Percentage of Patients Receiving RT	Percentage of Patients Receiving ChT/ITC	Major Findings	Toxicity of Radiotherapy
Broewer et al. [42]	Retrospective	124 (22)	Median 30 Gy/10 fx. (range 24–40 Gy)	54.5%	31.4%/7.4%	A complete course of WBRT was predictive of prolonged survival in a multivariate analysis (*p* = 0.019)	
Gani et al. [43]	Retrospective	27 (20)	Median 30 Gy/10 fx. (range 24–40 Gy)	100%	0%/0%	6-mo. OS 26%, 12-mo. OS 15% Median OS 2 mo. Improvement of neurological deficits: 11%	Grade 1 (erythema, alopecia, nausea, headache, fatigue—26% Grade 2 (tinnitus, alopecia, somnolence)—11.1% No grade 3 or 4 toxicity
Boogerd et al. [44]	Prospective, randomized	35 (35)	30 Gy/10 fx.	43%	46%/49%	WBRT with ITC is feasible and safe	DNL in one patient six mo. after WBRT and 3 patients without WBRT
Boogerd et al. [45]	Retrospective	14 (14)	(range 17.5–42 Gy)	29%	0%/100%	DNL occurred in 100% of irradiated patients and in 50% of patients without RT	DNL in all patients with WBRT and 50% of patients without WBRT
Okada et al. [46]	Retrospective	31 (31)	Median 30 Gy/10 fx. (range 20–37.5 Gy)	100%	0%/0%	Median OS for patients treated with 30 Gy in <10 fx. −0.6 mo. Median OS for patients treated with 30 Gy in ≥10 fx. −2.6 mo. (*p* < 0.1)	

BC, breast cancer; RT, radiotherapy; ChT, chemotherapy; ITC, intrathecal chemotherapy; Gy, Gray; fx., fractions; WBRT, whole-brain radiotherapy; mo., months; OS, overall survival; DNL, disseminated necrotizing leukoencephalopathy.

**Table 3 cancers-14-03899-t003:** Clinical trials evaluating the role of craniospinal irradiation for leptomeningeal carcinomatosis in breast cancer.

Authors	Number of Patients (BC)	Technique	Median Dose (Gy)/Fractions (N)	Clinical Response	Median OS (Mo.)	Grade 3–4 Toxicity (N)
Hermann et al. [51]	16 (9)	2D	36/20	68% improvement 12% stable	Entire group-2.8 CSI-1.84 CSI + ITC-3.7	Myelosuppression (5)
Harada et al. [52]	17 (6)	2D	41.4/23	70% improvement	8.8	Leukopenia (7) Thrombocytopenia (6) Fatigue, Nausea, Anorexia (4) Anemia (1) 1 toxic death
El Shafie et al. [53]	25 (15)	HTT	35.2/20	28% improvement 40% stable	4.8	Myelosuppression (8)
Devecka et al. [57]	19 (5)	2D until 2007, then HTT	30.6/19; boost to 37.6	58% improvement	34 (4.7 in BC)	Leukopenia (7) Thrombocytopenia (7) 1 toxic death-thrombosis
Schiopu et al. [58]	15 (6)	HTT	32.4/18	53% improvement (67% in BC)	3.0 (6.0 in BC)	Leukopenia (8) Thrombocytopenia (7) Anemia (5) Other (6) 3 toxic deaths (1 pulmonary embolism, 2 infections)

BC, breast cancer; Gy, gray; OS, overall survival; mo., months.; CSI, craniospinal irradiation; ITC, intrathecal chemotherapy; HTT, helical tomotherapy.

**Table 4 cancers-14-03899-t004:** Radiotherapy guidelines for leptomeningeal carcinomatosis.

Author	Title	Evidence	SRS	CSI	WBRT	IF-RT	Publication Date
The National Comprehensive Cancer Network (NCCN) [47]	NCCN Clinical Practice Guidelines in Oncology (NCCN Guidelines^®^) Central Nervous System Cancers	Expert consensus	Preferred option. Recommended in case of focal mass obstructing CSF flow	Not recommended due to high toxicity. Should be used only in highly selected patients (e.g., leukemia, lymphoma)	Preferred option	May be considered for palliation to neurologically symptomatic or painful sites (including spine and intracranial disease)	Version 2.2021—8 September 2021
European Association of Neuro-oncology (EANO)-European Society for Medical Oncology (ESMO) [24]	EANO–ESMO Clinical Practice Guidelines for diagnosis, treatment, and follow-up of patients with leptomeningeal metastasis from solid tumors	Expert consensus			Should be considered for extensive nodular or symptomatic linear LC	Should be considered for circumscribed, notably symptomatic lesions.	1 July 2017
The German Society of Radiation Oncology (DEGRO) [31,32]	DEGRO Practical Guidelines for Palliative Radiotherapy of Breast Cancer Patients: Brain Metastases and Leptomeningeal Carcinomatosis	Systematic review		Due to its myelotoxicity, should be considered only in selected cases, such as multiple circumscript plaques or nodules	Recommended for bulky disease or symptomatic regions	Recommended for bulky disease or symptomatic regions	26 January 2010

LC, leptomeningeal carcinomatosis; SRS, stereotactic radiosurgery; CSI, craniospinal irradiation; WBRT, whole-brain radiotherapy; IF-RT, involved-field radiotherapy; CSF, cerebrospinal fluid.

**Table 5 cancers-14-03899-t005:** Clinical trials involving radiotherapy for leptomeningeal carcinomatosis in breast cancer (ClinicalTrials.gov database).

ClinicalTrials.Gov Identifier	Recruitment Status	Intervention	Phase	Estimated Enrollment	Radiotherapy	RT Dose	RT Details	Primary Endpoint	Reference
NCT03719768	Active, not recruiting	Avelumab + RT	1	23	WBRT	30 Gy	Not reported	Safety and DLT	-
NCT03507244	Completed	IT pemetrexed + RT	1,2	34	WBRT, IF-RT	40 Gy/20 fx or 40–50 Gy/20–25 fx for spinal canal	Planning volume involves sites of symptomatic disease, bulky disease on MRI, WBRT and/or segment of the spinal canal	Incidence of treatment-related adverse events	[50]
NCT03520504	Active, not recruiting	Proton CSI	1	24	Proton	30 Gy (RBE)/10 fx or 25 Gy (RBE)/10 fx	Planning volume: brain, spinal cord, space containing CSF	Number of patients with DLT	-
NCT04192981	Recruiting	GDC-0084 + RT	1	36	WBRT	30 Gy/10fx	Not reported	MTD	-
NCT03082144	Completed	RT + IT MTX or RT + IT AraC	2	53	WBRT, IF-RT	40 Gy/20 fx or 40–50 Gy/20–25 fx for spinal canal	The sites of symptomatic disease, bulky disease at MRI, including the whole brain and cranial base and/or segment of spinal canal	Clinical response rate	-
NCT04588545	Recruiting	RT + IT trastuzumab/ pertuzumab	1,2	39	WBRT, IF-RT	30 Gy/10 fx or 20 Gy/5 fx	WBRT or focal brain/spine RT	Phase 1: MTD, Phase 2: OS	-
NCT04343573	Active, not recruiting	RT	2	111	Arm 1: Proton Arm 2: Photon	30 Gy/10 fx	Arm 1: Proton CSI Arm 2: Involved field photon RT including WBRT and/or focal spine RT	CNS progression-free survival	-
NCT04178343	Completed	RT	2	103	Tomotherapy	WBRT: 40 Gy/20 fx with SIB 60 Gy; WBRT: 50Gy/25 fx, depending on BM presence	Boost of the leptomeningeal metastases. WBRT of 50 Gy with hippocampus and brainstem sparing	OS	-
NCT00854867	Completed	WBRT + IT liposomal cytarabine	1	18	WBRT	38.4 Gy/20 fx	First two fx of 3 Gy, then 1.8 Gy/fx	The safety of WBRT concomitant with liposomal cytarabine	-
NCT05305885	Recruiting	IT pemetrexed +/− RT	Not applicable	100	WBRT, IF-RT	40 Gy/20 fx	Planning volume: sites of symptomatic disease, bulky disease on MRI, WBRT, and/or segment of the spinal canal	Clinical response rate	

RT, radiotherapy; WBRT, whole-brain radiotherapy; Gy, Gray; DLT, dose-limiting toxicity; IT, intrathecal; IF-RT, involved-field radiotherapy; fx, fractions; MRI, magnetic resonance imaging; CSI, craniospinal irradiation; RBE, relative biological effectiveness; CSF, craniospinal fluid; MTD, maximum tolerated dose; MTX, methotrexate; AraC, cytarabine; OS, overall survival; CNS, central nervous system; BM, brain metastases.

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
