# Peer review of "Radiotherapy for Leptomeningeal Carcinomatosis in Breast Cancer Patients: A Narrative Review"

_cancers, 2022, doi:10.3390/cancers14163899_

Round 1

Reviewer 1 Report

The review entitled "Radiotherapy for leptomeningeal carcinomatosis in breast cancer patients: a narrative review"presents the current knowledge of radiotherapy for the treatment of leptomeningeal carcinomatosis, an uncommon event in breast cancer. Despite the presence of various papers on this topic, none of them described the importance of radiotherapy. The authors present the impact of radiotherapy on survival, the whole grain radiotherapy, the stereotactic body radiation therapy, the proton therapy, the craniospinal irradiation and the guidelines for the radiotherapy. They also discuss the number of publications on radiotherapy for leptomeningeal carcinomatosis and also the clinical trials involving radiotherapy for leptomeningeal carcinomatosis. In this section the authors should add detailed datas on table 5 in order to better understand the results. The conclusions are consistent with the presented datas and arguments and the references are appropriate

Check the number of references (7 to 10)

Reviewer 2 Report

The manuscript is a review of the indications and contraindications in radiotherapy for the treatment of leptomeningeal carcinomatosis. This involvement occurs in a low percentage of patients with breast cancer, however it has a high associated mortality. I consider this review to be of great interest. Despite being a very well structured and very interesting manuscript, I have some comments.

Table 1: in the toxicity column. Does this toxicity only refer to that reported by radiotherapy? Please specify it.

Table 2: also include the toxicity column

Figure 1 and Figure 2 complete with data from 2021.

Review table format 5 page 13.

Throughout the manuscript many abbreviations are used: list or summarize all abbreviations.

I suggest completing the manuscript with a section on future perspectives in which the main lines of research in the treatment of leptomeningeal carcinomatosis are collected.
